# Community diversity is associated with intra-species genetic diversity and gene loss in the human gut microbiome

**Naïma Madi[1], Daisy Chen[2,3†], Richard Wolff[4†], B Jesse Shapiro[1,5,6,7,8]\*, Nandita R Garud[4,9]\***

[1]Département de sciences biologiques, Université de Montréal, Montréal, Canada; [2]Computational and Systems Biology, University of California, Los Angeles, Los Angeles, United States; [3]Bioinformatics and Systems Biology Program, University of California, San Diego, San Diego, United States; [4]Department of Ecology and Evolutionary Biology, University of California, Los Angeles, Los Angeles, United States; [5]McGill Genome Centre, McGill University, Montreal, Canada; [6]Quebec Centre for Biodiversity Science, Montreal, Canada; [7]McGill Centre for Microbiome Research, Montreal, Canada; [8]Department of Microbiology and Immunology, McGill University, Montreal, Canada; [9]Department of Human Genetics, University of California, Los Angeles, Los Angeles, United States

**\*For correspondence:**
jesse.shapiro@mcgill.ca (BJS);
ngarud@ucla.edu (NRG)

[†]These authors contributed
equally to this work

**Abstract** How the ecological process of community assembly interacts with intra-species diversity and evolutionary change is a longstanding question. Two contrasting hypotheses have been proposed: Diversity Begets Diversity (DBD), in which taxa tend to become more diverse in already diverse communities, and Ecological Controls (EC), in which higher community diversity impedes diversification. Previously, using 16S rRNA gene amplicon data across a range of microbiomes, we showed a generally positive relationship between taxa diversity and community diversity at higher taxonomic levels, consistent with the predictions of DBD (Madi et al., 2020). However, this positive 'diversity slope' plateaus at high levels of community diversity. Here we show that this general pattern holds at much finer genetic resolution, by analyzing intra-species strain and nucleotide variation in static and temporally sampled metagenomes from the human gut microbiome. Consistent with DBD, both intra-species polymorphism and strain number were positively correlated with community Shannon diversity. Shannon diversity is also predictive of increases in polymorphism over time scales up to ~4-6 months, after which the diversity slope flattens and becomes negative – consistent with DBD eventually giving way to EC. Finally, we show that higher community diversity predicts gene loss at a future time point. This observation is broadly consistent with the Black Queen Hypothesis, which posits that genes with functions provided by the community are less likely to be retained in a focal species' genome. Together, our results show that a mixture of DBD, EC, and Black Queen may operate simultaneously in the human gut microbiome, adding to a growing body of evidence that these eco-evolutionary processes are key drivers of biodiversity and ecosystem function.

## Editor's evaluation

This paper analyses meta-genomic human gut microbiome data to understand how biodiversity arises and can be maintained. It makes an important contribution by strengthening the diversity-begets-diversity hypothesis and linking it to signatures of gene loss expected from the Black Queen hypothesis. While only correlative data is used to draw conclusions, the methods are solid and alternative hypotheses are clearly outlined.

## Introduction

Our understanding of microbial evolution and diversification has been enriched by experimental studies of bacterial isolates in the laboratory, but it remains a challenge to study evolution in the context of more complex communities (*Lenski, 2017*). Ongoing advances in culture-independent technologies have allowed us to study bacteria in the complex and dense communities in which they naturally occur (*Garud and Pollard, 2020*). Within a community, individual players engage in many negative and positive ecological interactions. Negative interactions can originate from competition for resources and biomolecular warfare (*Hibbing et al., 2010*; *Mitri and Foster, 2013*), while positive interactions can stem from secreted metabolites that are used by other members of the community (cross-feeding) (*Venturelli et al., 2018*). These ecological interactions can create new niches and selective pressures, leading to eco-evolutionary feedbacks whose nature are yet to be fully understood.

Ecological interactions can yield positive or negative effects on the diversification of a focal species. Under the 'diversity begets diversity' (DBD) hypothesis, higher levels of community diversity increase the rate of speciation (or diversification, more generally) due to positive feedback mechanisms such as niche construction (*Calcagno et al., 2017*; *Schluter and Pennell, 2017*). Competition for limited niche space could also drive DBD if species diversify into new niches to avoid competition (*Meyer and Kassen, 2007*; *Mitri and Foster, 2013*; *Schluter, 2000*). By contrast, the 'ecological controls' (EC) hypothesis posits that competition for a limited number of niches at high levels of community diversity results in a negative effect on further diversification. Metabolic models predict that DBD may initially spur diversification due to cross-feeding, but the diversification rate eventually slows and reaches a plateau as metabolic niches are filled (*San Roman and Wagner, 2021*). These theoretical predictions are largely supported by our previous study involving 16S rRNA gene amplicon sequencing data from the Earth Microbiome Project, in which we observed a generally positive relationship (which we call the diversity slope; *Figure 1*) between community diversity and focal-taxon diversity at most taxonomic levels, reaching a plateau at the highest levels of diversity (*Madi et al., 2020*).

In this previous study, we found stronger support for DBD in the animal gut relative to more diverse microbiomes such as soils and sediments, which were closer to a plateau of diversity (*Madi et al., 2020*). While diversity slopes were generally positive at taxonomic levels as fine as amplicon sequence variants (akin to species or strains) within a genus, they were most positive at higher levels such as classes or phyla. A recent experiment on soil bacteria also found evidence of DBD at the family level, likely driven by niche construction and metabolic cross-feeding (*Estrela et al., 2022*). It therefore remains unclear if the predictions of DBD hold primarily at these higher taxonomic levels, involving the ecological process of community assembly, or if they also apply at the finer intra-species level. Within-host intra-species diversity can arise by co-colonization of a host by genetically distinct strains belonging to the same species or evolutionary diversification of a lineage via de novo mutation and gene gain/loss events within a host.

Such fine-scale strain-level variation has important functional and ecological consequences; among other things, strains are known to engage in interactions that cannot be predicted from their species identity alone (*Goyal et al., 2022*). Although closely-related bacteria are expected to have broadly similar niche preferences, finer-scale niches may differ below the species level (*Martiny et al., 2015*). For example, the acquisition of a carbohydrate-active enzyme by *Bacteroides plebeius* allows it to exploit a new dietary niche in the guts of people consuming nori (seaweed) (*Hehemann et al., 2010*), and single nucleotide adaptations permit *Enterococcus gallinarum* translocation across the intestinal barrier resulting in inflammation (*Yang et al., 2022*). Despite their potential phenotypic effects, it is unknown if such fine-scale genetic changes are favored by higher community diversity (due, for example, to niche construction, as predicted by DBD) or suppressed (due to competition for limited niche space, as predicted by EC). Competition could also lead to DBD if focal species evolve new niche preferences to avoid extinction (*Mitri and Foster, 2013*; *Schluter, 2000*) – an idea with some support in experimental microcosms (*Meyer and Kassen, 2007*) but largely unexplored in natural communities.

Here, we investigate the relationship between intra-species genetic diversity and community diversity in the human gut microbiome, a well-studied system in which we previously found support for DBD at higher taxonomic levels. We use static and temporal shotgun metagenomic data from a large panel of healthy adult hosts from the Human Microbiome Project (HMP) (*Lloyd-Price et al., 2017*; *Human Microbiome Project Consortium, 2012*) as well as from four healthy individuals sampled

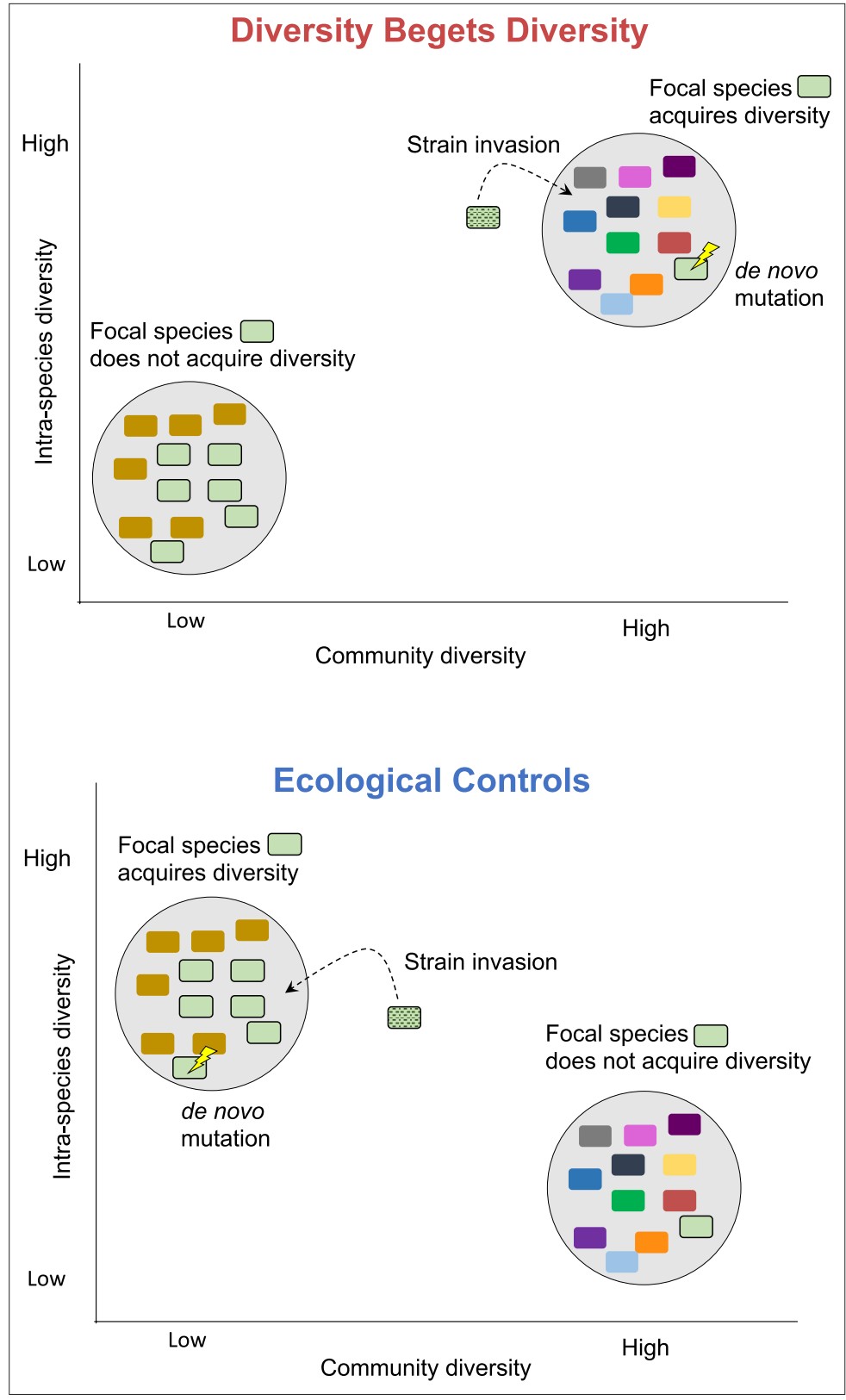

**Figure 1.** Diversity begets diversity (DBD) and ecological controls (EC) hypotheses illustrated. Hypothetical microbial communities are illustrated as gray circles containing assemblages of microbial species, shown in different colors. 'DBD' means that the focal species is more likely to acquire diversity – through de novo mutation, invasion of a different strain of the same species, or a combination of both – in a community with high diversity.

*Figure 1 continued on next page*

*Figure 1 continued*

This is because new niches are created in a more diverse community. By contrast, 'EC' means that the focal species is more likely to acquire diversity through strain invasion or mutation in a community with low diversity. This is because niches remain unfilled in a low-diversity community, while niche space is saturated in a high-diversity community, impeding further diversification.

almost daily over the course of 1 year (*Poyet et al., 2019*). Using metagenomic data allows us to track change in single nucleotide variation, strain diversity, and gene gain or loss events within relatively abundant species in the microbiome, and to study how these measures of intra-species diversity are associated with community diversity. Although such analyses of natural diversity cannot fully control for unmeasured confounding environmental factors, they are an important complement to controlled experimental and theoretical studies which lack real-world complexity.

## Results

We investigated the relationship between community diversity and within-species genetic diversity in human gut microbiota using two shotgun metagenomic datasets. First, we analyzed data from a panel of 249 healthy hosts (*Lloyd-Price et al., 2017*; *Human Microbiome Project Consortium, 2012*), in which stool samples were collected one to three times from each host at approximately 6-month intervals. Second, we analyzed data from four individuals sampled more densely over the course of ~18 months (*Poyet et al., 2019*). In both cases, we only consider intra-species diversity of relatively abundant species that are well sampled in these metagenomic datasets (Methods).

We examined several metrics of community diversity and intra-species diversity and calculated the slope of their relationship, defined as the diversity slope (*Figure 1*). We note that intra-species diversity can arise within hosts via de novo point mutation, gene gain or loss, or the coexistence of genetically distinct strains that diverged before colonizing the host. To quantify community diversity, we calculated Shannon diversity and richness at the species level. Shannon diversity is relatively insensitive to sampling effort (*Madi et al., 2020*; *Walters and Martiny, 2020*) but richness can be underestimated in low sample sizes. We therefore computed richness on data rarefied to an equal number of reads per sample, yielding generally similar results to unrarefied data (described below). In all cases, we included the number of reads per sample (coverage) as a covariate in our models, as this could affect estimates of both community diversity and intra-species diversity. To quantify intra-species diversity, we used a reference genome-based approach to call single nucleotide variants (SNVs) and gene copy number variants (CNVs) within each focal species and computed polymorphism rates, measured as the fraction of synonymous nucleotide sites in a species' core genome with intermediate allele frequencies (between 0.2 and 0.8) within a host (Methods). We also repeated the analysis on nonsynonymous sites, as these are subject to stronger selective constraints. As an additional metric of intra-species diversity, we inferred the number of strains within each species using StrainFinder applied to all polymorphic sites (including those outside the 0.2–0.8 frequency range) (*Smillie et al., 2018*).

### Community diversity is positively associated with intra-species polymorphism in the human gut microbiome

As an exploratory visualization, we began by plotting the relationship between community diversity and intra-species polymorphism rate calculated at synonymous sites in cross-sectional HMP metagenomes for the nine most prevalent species (*Figure 2A and B*). The slope of this relationship (the diversity slope; *Figure 1*) provides an indicator of the evidence for DBD (positive slope) or EC (flat or negative slope). The relationship between polymorphism rate and community diversity was mostly positive in the top nine most prevalent species in HMP hosts (*Figure 2A and B*). These nine species are used as a simple illustration of the diversity slope, not as a formal hypothesis-testing framework.

To generalize across species and to formally test the predictions of DBD, we fit generalized additive models (GAMs) to the HMP data. Using GAMs, we are able to model non-linear relationships and account for random variation in the strength of the diversity slope across bacterial species, the uneven number of samples per host, and the non-independence of samples from the same host (Methods; see *Supplementary file 1a* and *Supplementary file 2* section 1 for additional model details). These GAMs included 69 focal species with sufficient coverage to quantify within-species

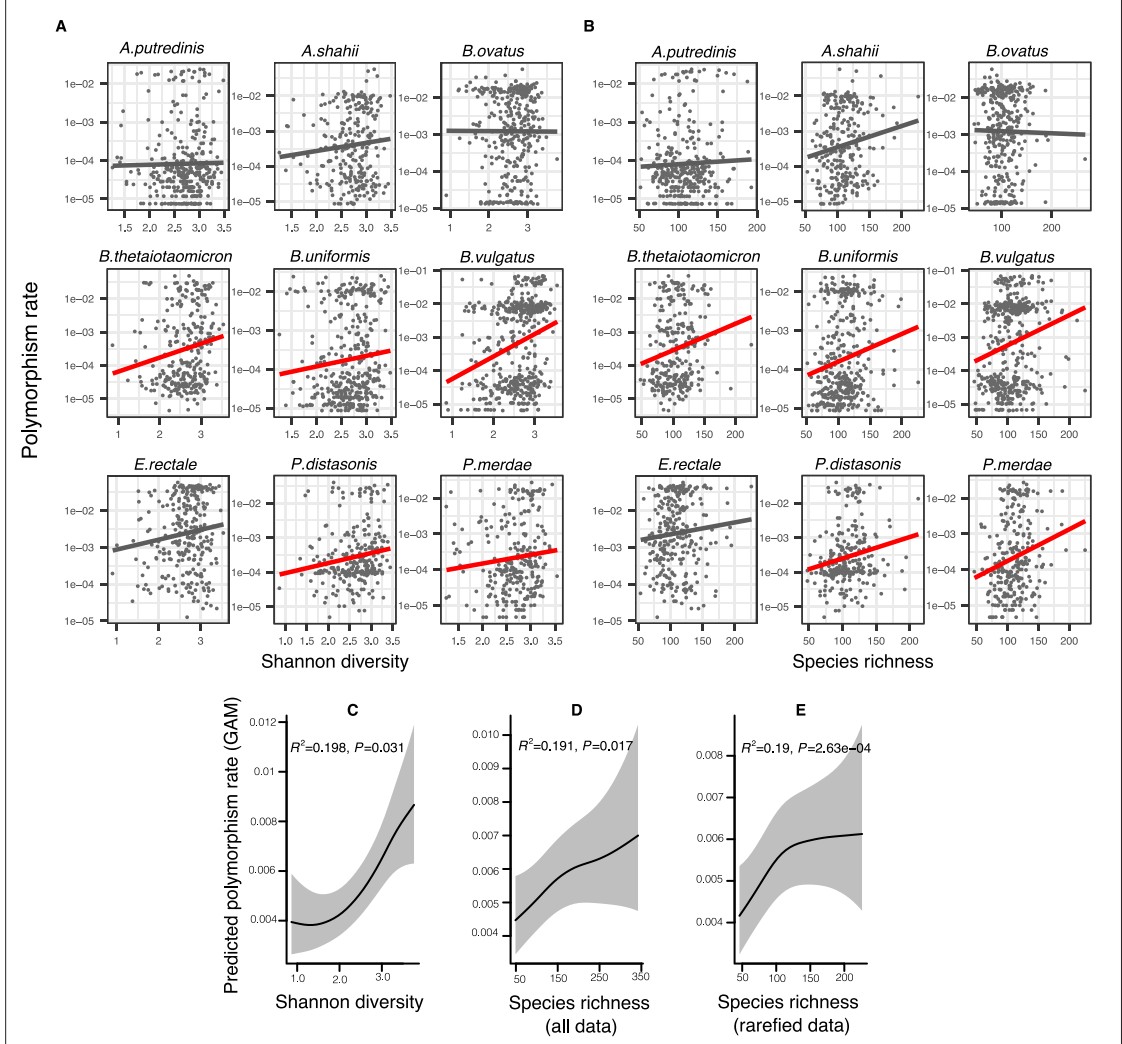

**Figure 2.** Positive association between community diversity and within-species polymorphism in cross-sectional Human Microbiome Project (HMP) samples. (**A**) Scatter plots showing the relationship between community Shannon diversity and within-species polymorphism rate (estimated at synonymous sites) in the nine most prevalent species in HMP. (**B**) Scatter plots showing the relationship between species richness and within-species polymorphism rate in the nine most prevalent species in HMP. These are simple correlations to show the relationships in the raw data. Significant correlations are shown with red trendlines (Spearman correlation, p<0.05); non-significant trendlines are in gray. Results of generalized additive models (GAMs) predicting polymorphism rate in a focal species as a function of (**C**) Shannon diversity, (**D**) species richness estimated on all sequence data, and (**E**) species richness estimated on rarefied sequence data. GAMs are based on data from 69 bacterial species across 249 HMP stool donors. Adjusted $R^2$ and Chi-square p-values corresponding to the predictor effect are displayed in each panel. Shaded areas show the 95% confidence interval of each model prediction. See ***Supplementary file 1a*** and ***Supplementary file 2*** section 1 for detailed model outputs.

The online version of this article includes the following figure supplement(s) for figure 2:

**Figure supplement 1.** Results of generalized additive models predicting within-species polymorphism rate (at synonymous sites) as a function of community diversity at higher taxonomic levels (Human Microbiome Project [HMP] data).

**Figure supplement 2.** Results of generalized additive models predicting within-species polymorphism rate (at nonsynonymous sites) in a focal species as a function of community diversity at higher taxonomic levels (Human Microbiome Project [HMP] data).

polymorphism (Methods); the results therefore apply to relatively abundant species in the human gut microbiome. GAMs showed an overall positive association between within-species polymorphism and Shannon diversity (***Figure 2C***, GAM, p=0.031, Chi-square test) as well as between within-species polymorphism and community richness after controlling for coverage as a covariate (***Figure 2D***, GAM, p=0.017, Chi-square test) or rarefying samples to an equal number of reads (***Figure 2E***, GAM, p=2.63e-04, Chi-square test). The random effect of species identity is highly significant in all models, indicating that each bacterial species has its own characteristic diversity slope (***Supplementary***

*file 1a*). It appears that synonymous polymorphism reaches a plateau at high levels of community richness, which is particularly evident when using rarefied data (*Figure 2E*). Using the same GAMs applied to nonsynonymous polymorphism, we found no significant associations between diversity and within-species polymorphism rate (GAM, p>0.05, Chi-square test) (*Supplementary file 1b*, *Supplementary file 2* section 4). This could be due to lower statistical power, since there are fewer nonsynonymous than synonymous sites, or it could reflect a true difference in the diversity slope between these site categories.

These generally positive correlations between focal species polymorphism and species-level measures of community diversity also hold when community diversity is measured at higher taxonomic levels; specifically, synonymous polymorphism rate was significantly positively associated with Shannon diversity calculated at the genus and family levels (GAMs, p<0.05, Chi-square test) (*Figure 2—figure supplement 1*, *Supplementary file 1c*). However, synonymous polymorphism rate was not significantly associated with Shannon diversity calculated at the highest taxonomic levels (order, class, and phylum, GAMs, p>0.05, Chi-square test). The positive correlation between polymorphism rate and richness held at all taxonomic levels (GAMs, p<0.05, Chi-square test) (*Figure 2—figure supplement 1*, *Supplementary file 1c*, *Supplementary file 2* sections 2 and 3). When estimated at nonsynonymous sites, polymorphism rate was not significantly correlated with Shannon diversity at any taxonomic level (GAMs, p>0.05, Chi-square test), but was positively correlated with richness at the highest levels (phyla, class, and order, p=3e-04, p=0.017, and p=6.11e-04, respectively, Chi-square test from GAMs) (*Figure 2—figure supplement 2*, *Supplementary file 1d*, *Supplementary file 2* sections 5 and 6). Even when not statistically significant, the diversity slopes were generally positive at all taxonomic levels for both synonymous and nonsynonymous polymorphism (*Figure 2—figure supplements 1 and 2*). Overall, these results are consistent with the predictions of DBD at most taxonomic levels. However, slightly different relationships are observed when considering different measures of community diversity (Shannon or richness) and different components of within-species diversity (nonsynonymous or synonymous).

## Different measures of community diversity have contrasting associations with intra-species strain diversity

Within host polymorphism rates span several orders of magnitude ($10^{-5}$/bp to $10^{-2}$/bp), largely due to the fact that strain content is variable across hosts. As previously argued (*Garud et al., 2019*), with conservatively high estimates for mutation rate ($\mu \sim 10^{-9}$) (*Sung et al., 2012*), generation times ($\sim 10$/ day) (*Poulsen et al., 1995*), and time since colonization (<100 years), polymorphism rates of $\sim 10^{-2}$/bp or more are inconsistent with within-host diversification of a single colonizing lineage. Therefore, hosts with relatively high intra-host polymorphism rates are likely colonized by mixtures of multiple strains that diverged long before colonizing a host. Moreover, recent work suggests that the numbers and genetic composition of strains colonizing a host can vary from host to host (*Garud et al., 2019*; *Olm et al., 2017*; *Russell and Cavanaugh, 2017*; *Truong et al., 2017*; *Verster et al., 2017*). The associations between polymorphism and community diversity (*Figure 2*) are likely driven by a combination of de novo mutation and co-colonization by multiple strains.

To separate these two sources of diversity and to explicitly account for the strain structure within hosts, we inferred the number of strains per focal species with StrainFinder (*Smillie et al., 2018*) (Methods) and used strain number as another quantifier of intra-species diversity. The relationship between community diversity and strain number varied depending on the focal species and the measure of community diversity. For example, the inferred number of *Bacteroides vulgatus* strains increased with community diversity, while *Bacteroides uniformis* strain count decreased or remained flat (*Figure 3A and B*). Expanding upon these examples, we used generalized linear mixed models (GLMMs) to investigate the relationship between the number of strains per focal species and community diversity, while taking into account coverage per sample as a covariate and variation between species, hosts and samples as random effects (Methods). GLMMs are a special case of GAMs that can handle overdispersed, zero-truncated count data such as strain counts. The number of strains per focal species was positively correlated with community Shannon diversity (GLMM, p=3.58e-07, likelihood ratio test [LRT]) (*Figure 3C*, *Supplementary file 1e*, *Supplementary file 2* section 7.1). This suggests that the positive correlation between polymorphism rate and Shannon diversity (*Figure 2*) is due at least in part to strain diversity.

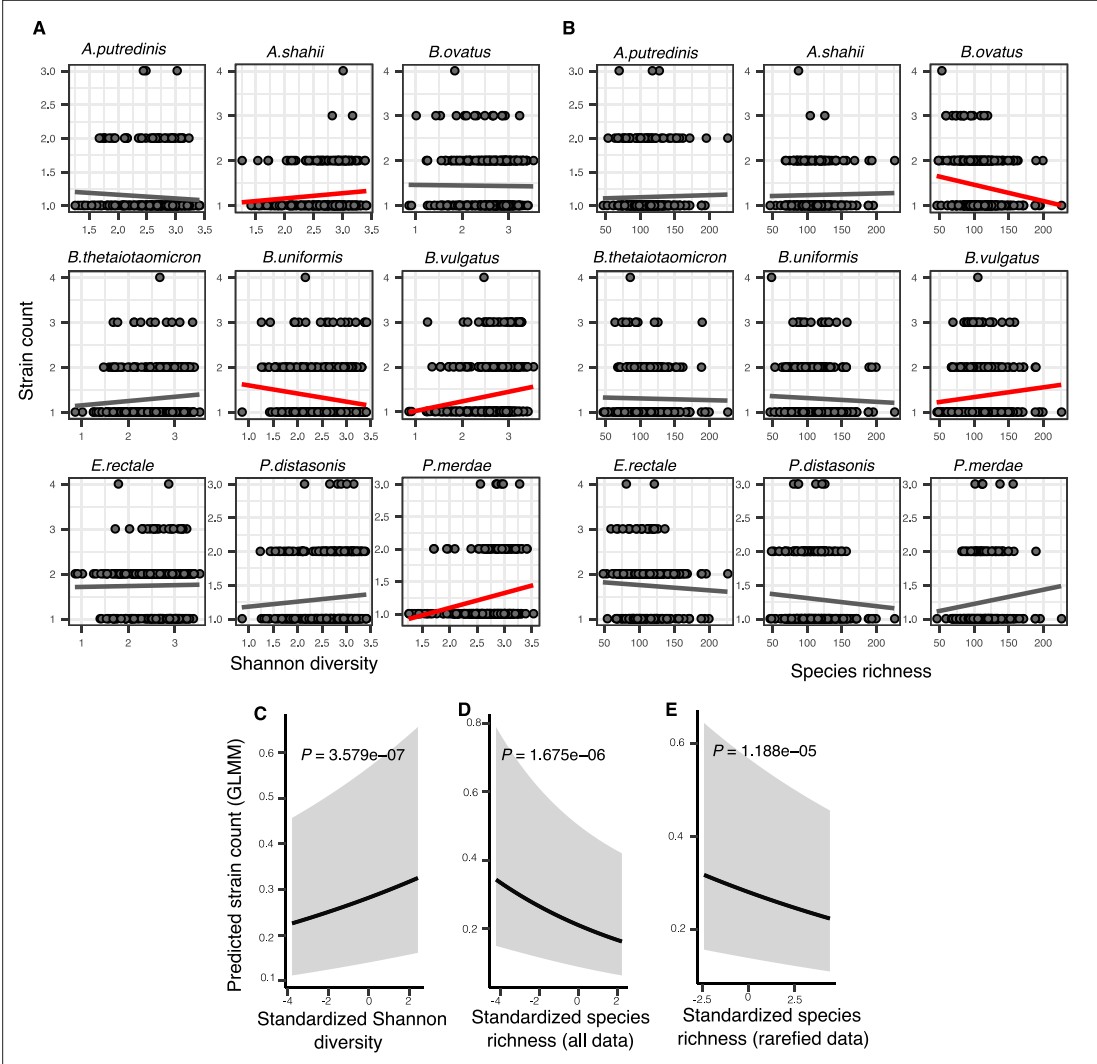

**Figure 3.** Associations between community diversity and strain number in cross-sectional Human Microbiome Project (HMP) samples. (**A**) Scatter plots showing the relationship between Shannon diversity and the inferred number of strains within each of the nine most prevalent species in HMP. (**B**) Scatter plots showing the relationship between species richness and the inferred number of strains within each of the nine most prevalent species in HMP. Significant linear correlations are shown with red trendlines (Pearson correlation, p<0.05); non-significant trend lines are in gray. Results of generalized linear mixed models (GLMMs) predicting strain count in a focal species as a function of (**C**) Shannon diversity, (**D**) species richness estimated on all data, and (**E**) species richness estimated on rarefied sequence data. Diversity estimates (X-axis) are standardized to zero mean and unit variance in the models. The Y-axis shows the mean number of strains per focal species predicted by the GLMM. GLMMs are based on data from 184 bacterial species across 249 HMP stool donors. p-Values (likelihood ratio test) are displayed in each panel. Shaded areas show the 95% confidence interval of each model prediction. See *Supplementary file 1e* and *Supplementary file 2* section 7 for detailed model outputs.

The online version of this article includes the following figure supplement(s) for figure 3:

**Figure supplement 1.** Results of generalized linear mixed models predicting strain count in a focal species as a function of community diversity at higher taxonomic levels (Human Microbiome Project [HMP] data).

By contrast, species richness was negatively correlated with strain number (GLMM, p=1.67e-06, LRT) (*Figure 3D*, *Supplementary file 1e*, *Supplementary file 2* section 7.2). The negative relationship with richness was unlikely to be confounded by sequencing depth, since the same result was obtained using rarefied data (*Figure 3E*, *Supplementary file 1e*, *Supplementary file 2* section 7.3). The negative strain number-richness relationship also held at all other taxonomic ranks (GLMM, p<0.05, LRT), while the strain number-Shannon diversity relationship was generally positive (*Figure 3—figure supplement 1*, *Supplementary file 1f*, *Supplementary file 2* sections 8 and 9). These effects also appear to be species-specific: for example, the number of *B. vulgatus* strains per host is positively

correlated with both Shannon diversity and richness (consistent with DBD predictions) whereas *Bacteroides ovatus* has no relationship with Shannon diversity but a negative correlation with richness (consistent with EC; *Figure 2A and B*). Together, these results reveal that different components of community diversity can have contrasting effects on the diversity slope.

## Community Shannon diversity is a predictor of intra-species polymorphism and gene loss in time series data

Our analyses thus far have considered only individual time points, which represent static snapshots of the dynamic processes of community assembly and evolution in the microbiome. To interrogate these phenomena over time, we analyzed 160 HMP subjects who were sampled two to three times ~6 months apart. Under a DBD model, we expect community diversity at an earlier time point to result in higher within-species polymorphism at a future time point. To test this expectation, we defined 'polymorphism change' as the difference between polymorphism rates at the two time points (Methods). We also investigated the effects of community diversity on gene loss and gain events within a focal species, as such changes in gene content are known to occur frequently within host gut microbiomes (*Garud et al., 2019*; *Groussin et al., 2021*; *Yaffe and Relman, 2020*; *Zhao et al., 2019*). Here, a gene was considered absent if its copy number ($c$) was <0.05 and present if $0.6 \le c \le 1.2$. As in the cross-sectional analyses above, we also controlled for sequencing depth of the sample and excluded genes with aberrant coverage or presence in multiple species (Methods).

In HMP samples, polymorphism change showed no significant relationships with community diversity at the earlier time point, whether it was estimated with Shannon index or species richness (GAM, p>0.05) (*Supplementary file 2* section 10.1). These results suggest that DBD is negligible or undetectable over ~6 month time lags in the human gut. By contrast, we found that gene loss in a focal species between two consecutive time points was positively correlated with community diversity at the earlier time point (*Figure 4*; GLMM, p=0.028, p=0.034, and p=0.049, LRT for Shannon, richness and rarefied richness, respectively) (*Supplementary file 1g*, *Supplementary file 2* section 10.3). Gene gains did not show any significant relationships with community diversity (GLMM, p>0.05). Selection for gene loss in more diverse communities is a prediction of the Black Queen hypothesis (BQH), provided that higher community diversity results in more redundant gene functions that compensate for losses in a focal species (*Morris et al., 2012*). Most species in HMP samples lost fewer than 10 genes over ~6 months – consistent with de novo deletion events of a few genes – but occasionally hundreds of genes were lost from a host, suggesting that strains with smaller genomes were selected in more diverse communities (*Figure 4A and B*).

To study these dynamics at higher temporal resolution, we analyzed shotgun metagenomic data from four more frequently sampled healthy individuals from a previous study (*Poyet et al., 2019*). Stool from donor *am* was sequenced over 18 months with a median of 1 day between samples; *an* over 12 months (median 2 days between samples); *ao* over 5 months (median 1 day between samples); and *ae* over 7 months (median 2 days between samples). In this data, we tracked both polymorphism change and gene gains and losses between two successive time points in 15 species with a minimal marker gene coverage of 10 in at least 10 samples. These include seven species of *Bacteroides*, two *Eubacterium,* two *Faecalibacterium*, two *Ruminococcus,* as well as *Alistipes putredinis* and *Parabacteroides merdae*.

Using the Poyet dataset, we asked whether community diversity in the gut microbiome at one time point could predict polymorphism change at a future time point by fitting GAMs with the change in polymorphism rate as a function of the interaction between community diversity at the first time point and the number of days between the two time points. Shannon diversity at the earlier time point was correlated with increases in polymorphism (consistent with DBD) up to ~150 days (~4.5 months) into the future (*Figure 5—figure supplement 1*), but this relationship became weaker and then inverted (consistent with EC) at longer time lags (*Figure 5A*, *Supplementary file 1h*, GAM, p=0.023, Chi-square test). The diversity slope is approximately flat for time lags between 4 and 6 months, which could explain why no significant relationship was found in HMP, where samples were collected every ~6 months. No relationship was observed between community richness and changes in polymorphism (*Supplementary file 1h*, GAM, p>0.05).

We next asked if community diversity at one time point could predict gene gains or losses at future time points by fitting GLMMs (analogous to the GAMs above, but more appropriate for gain/

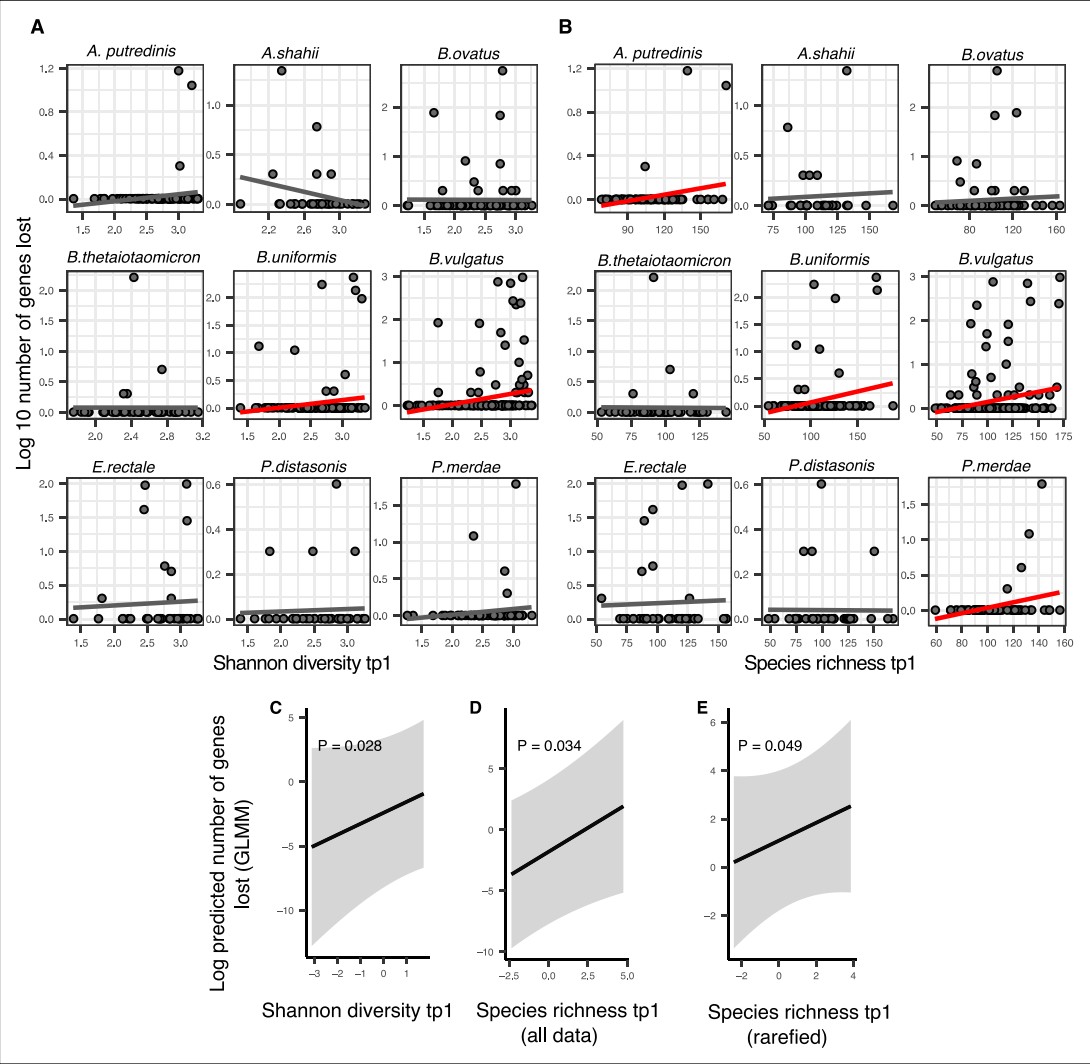

**Figure 4.** Positive association between community diversity and gene loss in Human Microbiome Project (HMP) time series. (**A**) Scatter plots showing the relationship between Shannon diversity at time point 1 (tp1) and gene loss between tp1 and tp2 within each of the nine most prevalent species in HMP. (**B**) Scatter plots showing the relationship between species richness at tp1 and gene loss between tp1 and tp2 within each of the nine most prevalent species in HMP. Significant linear correlations are shown with red trendlines (Pearson correlation, p<0.05); non-significant trend lines are in gray. The Y-axis is plotted on a log10 scale for clarity. Results of generalized linear mixed models (GLMMs) predicting gene loss in a focal species as a function of (**C**) Shannon diversity, (**D**) species richness estimated on all data, and (**E**) species richness estimated on rarefied sequence data. p-Values (likelihood ratio test) are displayed in each panel. Shaded areas show the 95% confidence interval of each model prediction. The Y-axis is plotted on the link scale, which corresponds to log for negative binomial GLMMs with a count response. GLMMs are based on data from 54 bacterial species across 154 HMP stool donors sampled at more than one time point. See *Supplementary file 1g* and *Supplementary file 2* section 10 for detailed model outputs.

loss count data). Our method does not explicitly distinguish between gene gain/loss arising from recombination or deletion versus replacement of strains with different gene content. We found that community Shannon diversity predicted future gene loss in a focal species, and this effect became stronger with longer time lags (*Figure 5B*, *Supplementary file 1i*, GLMM, p=0.006, LRT for the effect of the interaction between the initial Shannon diversity and time lag on the number of genes lost). The model predicts that increasing Shannon diversity from its minimum to its maximum would result in the loss of 0.075 genes from a focal species after 250 days. In other words, about one of the 15 focal species considered would be expected to lose a gene in this time frame.

Higher Shannon diversity was also associated with fewer gene gains, and this relationship also became stronger over time (*Figure 5C*, *Supplementary file 1i*, GLMM, p=1.11e-09, LRT). We found a similar relationship between community species richness and gene gains, although the relationship

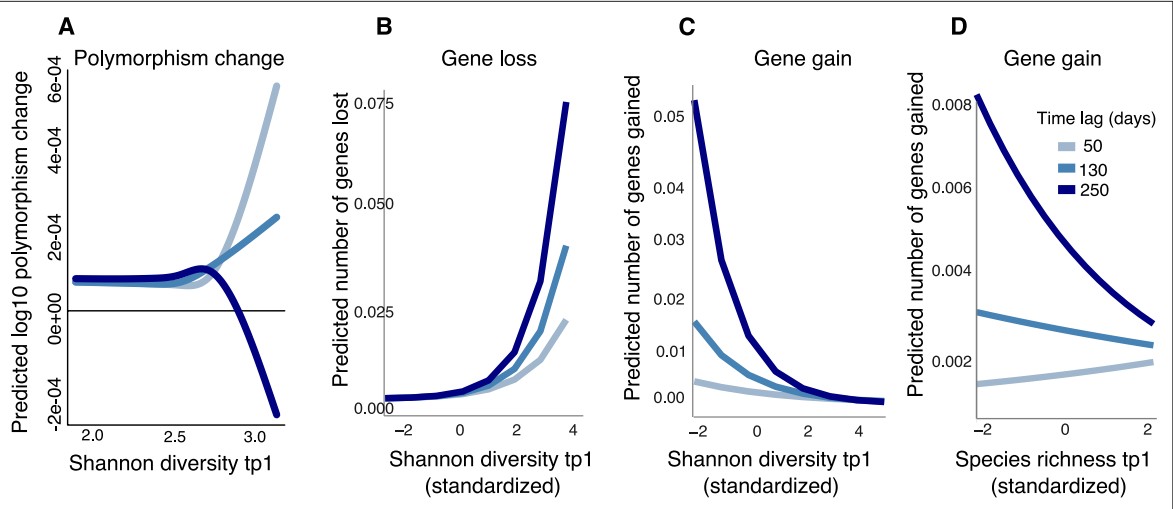

**Figure 5.** Community diversity is associated with increases in focal species polymorphism over short time lags and net gene loss in dense gut microbiome time series. (**A**) Results of a generalized additive model (GAM) predicting polymorphism change in a focal species as a function of the interaction between Shannon diversity at the first time point and the time lag (days) between two time points in data from Poyet et al. The response (Y-axis) was log-transformed in the Gaussian GAM. Results of generalized linear mixed models (GLMMs) predicting (**B**) number of genes lost and (**C**) number of genes gained between two time points in a focal species as a function of the interaction between Shannon diversity at the first time point and the time lag between the two time points. (**D**) Results of the GLMM predicting the number of genes gained in a focal species as a function of the interaction between rarefied species richness at the first time point and the time lag between the two time points. The illustrated time lags correspond to the first quartile (50 days), the median (130 days), and the third quartile (250 days). See *Supplementary file 1h and i* and *Supplementary file 2* section 11 for detailed model outputs. These analyses are based on data from 15 bacterial species across four stool donors from Poyet et al. Only statistically significant relationships are plotted. Non-significant relationships are not shown: the GAM predicting polymorphism change as a function of rarefied richness (p>0.05) and the GLMM predicting the number of genes lost as a function of rarefied richness (p>0.05).

The online version of this article includes the following figure supplement(s) for figure 5:

**Figure supplement 1.** Results of a generalized additive model (GAM) predicting polymorphism change in a focal species as a function of the interaction between Shannon diversity at the first time point and the time lag (days) between two time points in the Poyet time series.

was slightly positive at shorter time lags (*Figure 5D*, *Supplementary file 1i*, GLMM, p=3.41e-04, LRT). No significant relationship was observed between richness and gene loss (*Supplementary file 1i*, GLMM, p>0.05). Taken together with the HMP results (*Figure 4*), these longer time series reveal how the sign of the diversity slope can vary over time and how community diversity is generally predictive of reduced focal species gene content.

## Discussion

How eco-evolutionary feedbacks shape biological communities is an open question that, to date, has received substantial experimental and theoretical attention but is challenging to address in nature. In our previous study using 16S rRNA amplicon sequences from the Earth Microbiome Project, we found generally positive diversity slopes that eventually flattened at high levels of community diversity (*Madi et al., 2020*). This pattern is generally consistent with the predictions of DBD during the early stages of community assembly, but at later stages becomes more consistent with EC as niches become filled. Based on the time series metagenomic data analyzed here, the predictions of DBD also tend to hold over short time scales but fail over longer time scales of several months. Whether this leads to a terminal plateau of diversity, or whether ecological disturbances lead to cycles of DBD and EC, deserves further study.

In our previous study, the animal gut microbiome had one of the highest positive diversity slopes, making it an ideal candidate for investigating eco-evolutionary interactions at greater intra-species resolution using metagenomic data. In this follow-up study, we investigate the same phenomenon at a subspecies level, with results that are broadly consistent with the predictions of DBD giving way to EC over long time scales. We note that experiments supporting DBD have generally been conducted over short time scales ranging from 2 to 20 days (*Estrela et al., 2022*; *Jousset et al., 2016*), consistent with

the importance of DBD early in community assembly. We also identify several nuances and caveats to this general conclusion, which are discussed below in detail.

Another recent study also found evidence for eco-evolutionary feedbacks in the HMP, in the form of a positive relationship between evolutionary modifications or strain replacements in a focal species and community diversity (*Good and Rosenfeld, 2022*). Using a model, they further showed that these eco-evolutionary dynamics could be explained by resource competition and did not require the cross-feeding interactions previously invoked (*Estrela et al., 2022*; *San Roman and Wagner, 2021*; *San Roman and Wagner, 2018*) to explain DBD at higher taxonomic levels. This could be because cross-feeding operates at the family or genus level and is less relevant at finer evolutionary scales.

There are several noteworthy caveats to our study. First, using metagenomic data from human microbiomes allowed us to study genetic diversity, but limited us to considering only relatively abundant species with genomes that were well covered by short sequence reads. Deeper or more targeted sequencing may permit us to determine whether the same patterns hold for rarer members of the microbiome. However, it is notable that the majority of the dozens of species across the two datasets analyzed support DBD, suggesting that the phenomenon may generalize.

Second, we cannot establish causal relationships without controlled experiments. We are therefore careful to conclude that positive diversity slopes are consistent with the predictions of DBD, and negative slopes with EC, but unmeasured environmental drivers could be at play. For example, increased dietary diversity could simultaneously select for higher community diversity and also higher intra-species diversity. In our previous study, we found that positive diversity slopes persisted even after controlling for potential abiotic drivers such as pH and temperature (*Madi et al., 2020*), but a similar analysis was not possible here due to a lack of metadata. Neutral processes can account for several ecological patterns such as species-area relationships (*Hubbell, 2001*), and must be rejected in favor of niche-centric models like DBD or EC. Using neutral models without DBD or EC, we found generally flat or negative diversity slopes due to sampling processes alone and that positive slopes were hard to explain with a neutral model (*Madi et al., 2020*). These models were intended mainly for 16S rRNA gene sequence data, but we expect the general conclusions to extend to metagenomic data. Nevertheless, further modeling and experimental work will be required to fully exclude a neutral explanation for the diversity slopes we report in the human gut microbiome.

Based on controlled experiments (*Estrela et al., 2022*) and modeling studies (*San Roman and Wagner, 2021*), DBD is a plausible causal explanation for positive diversity slopes in the gut microbiome. Although they also note that causality is difficult to establish, *Good and Rosenfeld, 2022* suggest the importance of focal species evolution as a driver of changes in community structure, as shown in an experimental study of *Pseudomonas* in compost communities (*Padfield et al., 2020*). Clearly, further work is needed to establish the extent and relative rates of eco-evolutionary feedback in both directions. How these feedbacks among bacteria are influenced by abiotic factors and by interactions with fungi, archaea, and phages also deserves further study.

Third, the diversity slope changes depending on which component of within-species diversity or community diversity is considered. Notably, the number of strains within a focal species is positively correlated with Shannon diversity, but inversely correlated with species richness, suggesting that the ability of strains to colonize a host may be associated with higher community evenness rather than total species count. Higher evenness might maximize the chance of inter-species interactions, whereas higher richness might be driven by rare species that are less likely to interact. Although Shannon diversity is considered to be more robust and informative than richness in estimating bacterial diversity (*He et al., 2013*; *Reese and Dunn, 2018*), we observe the same contrasting results between Shannon diversity and richness when community diversity is calculated at higher taxonomic levels, suggesting that this pattern is not due to artifacts such as sequencing effort.

Our measures of intra-species diversity included both synonymous and nonsynonymous SNVs, inferred strain richness, and gene content. Synonymous nucleotide variation was consistently and positively associated with both community richness and Shannon diversity at all taxonomic levels (although not always with statistical significance). Nonsynonymous variation also tended to track positively with both measures of community diversity but was only statistically significantly associated with phylum and class richness. This suggests that evolutionarily older, less selectively constrained synonymous mutations and more recent nonsynonymous mutations that affect protein structure both track similarly with measures of community diversity. Nonetheless, a parsimonious explanation for

possible differences between the two classes is that while they are affected similarly, we have more statistical power to identify correlations in the more numerous synonymous mutations. This merits further investigation.

Metagenomes from the same individual sampled over time allowed us to detect gene gain and loss events. In both HMP and Poyet et al. time series, community diversity was predictive of future gene loss in a focal species. This phenomenon is not explicitly predicted by either DBD or EC but it is compatible with aspects of the BQH, with some caveats. BQH predicts that a focal species will be less likely to encode genes with functions provided by other members of the surrounding community if such functions are 'leaky' and available as diffusible public goods (*Morris et al., 2012*). The BQH could also act as a driver of polymorphism within a species (*Morris et al., 2014*). Gene loss may be adaptive, provided that there is a cost to encoding and expressing the relevant genes (*Albalat and Cañestro, 2016*; *Koskiniemi et al., 2012*; *Simonsen, 2022*). The tendency for reductive genome evolution in bacteria is well established (*Albalat and Cañestro, 2016*; *Koskiniemi et al., 2012*; *Puigbò et al., 2014*). Genome reduction is a particular hallmark of endosymbiotic bacteria, which depend on their hosts for many metabolic gene products (*McCutcheon and Moran, 2011*; *Nikoh et al., 2011*). It has been shown that uncultivated bacteria from the gut have undergone considerable genome reduction, which may be an adaptive process that results from reliance on public goods (*Nayfach et al., 2019*). In the gut microbiome, the BQH has been invoked to explain the distribution of genes involved in vitamin B metabolism (*Sharma et al., 2019*) and iron acquisition (*Vatanen et al., 2019*).

Our findings in human gut metagenomes are compatible with the BQH under the assumption that increasing community diversity also increases the availability of leaky gene products – which may not be the case if genomes in the gut microbiome are functionally redundant, as inferred in a recent study (*Tian et al., 2020*). This study found that species in the gut microbiome were highly redundant at the level of annotated metabolic pathways (KEGG orthologs) and that more functionally redundant microbiomes were more resistant to colonization by fecal transplants. Relatively low redundancy microbiomes could therefore be more easily colonized but might also require migrants to encode more gene functions in order to persist. Importantly, functional redundancy may be high at the level of well-annotated metabolic functions, but low at the finer level of individual gene families, as demonstrated in marine microbiomes (*Galand et al., 2018*) but not yet studied explicitly in the gut. Here, we report that genome reduction in the gut is higher in more diverse gut communities. This could be due to de novo gene loss, preferential establishment of migrant strains encoding fewer genes, or a combination of the two. The mechanisms underlying this correlation remain unclear and could be due to biotic interactions – including metabolic cross-feeding as posited by some models (*Estrela et al., 2022*; *San Roman and Wagner, 2021*; *San Roman and Wagner, 2018*) but not others (*Good and Rosenfeld, 2022*) – or due to unknown abiotic drivers of both community diversity and gene loss. Finally, we measured community diversity from the phylum to the species level, not below. We therefore did not investigate how the BQH could extend to maintain gene content variation within a species, as has been shown experimentally in *Escherichia coli* (*Morris et al., 2014*). This could be an avenue for future work.

In our previous analysis of lower-resolution 16S rRNA amplicon sequences, we reported a tendency for focal genera with larger genomes to have higher diversity slopes, perhaps because they experience stronger DBD (*Madi et al., 2020*). At face value, this tendency seems at odds with the BQH, which predicts genome reduction in more diverse communities. This apparent contradiction may be reconciled by considering eco-evolutionary dynamics on different time scales. A recent study used phylogenetic and metabolic reconstructions to show that gene gains often drive metabolic dependencies among bacteria (*Goyal, 2021*), potentially explaining why genera with larger maximum genome size could experience stronger DBD. Our earlier study only had the genetic resolution to consider focal taxa down to the genus level, and by using the maximum genome size observed in a public database we did not capture the dynamic process of gene gain and loss within a species, as was possible in the current metagenomic study. It is therefore possible that on longer (ecological) time scales, larger genomes have more metabolic interactions and thus experience stronger DBD, while genome reduction in more diverse communities occurs on shorter (evolutionary) time scales.

In summary, we demonstrate how metagenomic data can be used to test the predictions of eco-evolutionary theory, including DBD, EC, and the BQH. It remains to be seen whether the distinct eco-evolutionary processes proposed by DBD and the BQH operate orthogonally or whether they

interact. If BQH leads to gene losses that remain polymorphic rather than being lost entirely from the species (*Morris et al., 2014*) – or invasions of strains with fewer genes that remain incomplete and do not replace the resident strain – this could be viewed as a form of diversification and perhaps a special case of DBD. Here, we considered gene loss as a directional process; we did not attempt to distinguish between directional changes in gene copy number and the complete extinction of a gene, which is difficult to show using metagenomic data. Future work could attempt to resolve this point and to potentially combine DBD and BQH into a unified theory.

## Methods

### Metagenomic analyses

#### Estimation of species, gene, and SNV content of metagenomic samples

We used MIDAS (Metagenomic Intra-Species Diversity Analysis System, version 1.2, downloaded on November 21, 2016) (*Nayfach et al., 2016*) to estimate within-species nucleotide and gene content of raw metagenomic whole genome shotgun sequencing data for HMP1-2 and *Poyet et al., 2019*, data. MIDAS relies on a reference database comprised of 31,007 bacterial genomes that are clustered into 5952 species, covering roughly 50% of species found in human stool metagenomes from 'urban' individuals. Described below are the parameters used to estimate species abundances, SNVs, and gene CNVs with MIDAS.

#### Estimation of species content

We estimated species abundances, SNVs, and CNVs by mapping metagenomic shotgun reads to reference genomes. Since a component of this work relies on quantifying polymorphism and CNV changes over time, we constructed a 'personal' reference database to avoid spurious inferences of allele frequency and CNV changes due to errors in mapping of reads to regions of the genome shared by multiple species (*Garud et al., 2019*). This per-host reference database was comprised of the union of all species present at one or more time points so as to be as inclusive as possible to prevent reads from being 'donated' to reference genome, while also being selective to prevent a reference genome from 'stealing' reads from a species truly present.

To estimate the species relative abundances for each host × time point sample, we mapped reads to 15 universal single-copy marker genes that are a part of the MIDAS pipeline (*Nayfach et al., 2016*; *Wu et al., 2013*) and belong to the 5952 species in the MIDAS reference database. A species with an average marker gene coverage ≥3 was considered present for the purposes of building a per-host database for mapping reads to infer SNVs and CNVs below. The per-host database was constructed by including all species present at one or more time points with coverage ≥3. However, more stringent thresholds were imposed for calling SNVs and CNVs, as described below.

#### Estimation of CNVs

To estimate gene CNVs, we mapped reads to the pangenomes of species present in a host's personal database using Bowtie2 (*Langmead and Salzberg, 2012*) with default MIDAS settings (local alignment, MAPID ≥94.0%, READQ ≥20, and ALN_COV ≥0.75). Each gene's coverage was estimated by dividing the total number of reads mapped to a given gene by the gene length. These genes included the aforementioned 15 universal single-copy marker genes. A given gene's copy number ($c$) was estimated by taking the ratio of its coverage and the median coverage of the species' single-copy marker genes.

With these copy number values, we estimated the prevalence of genes in the between-host population, defined as the fraction of samples with copy number $c \leq 3$ and $c \geq 0.3$ (conditional on the mean single gene marker coverage being ≥5×). For each species, we computed 'core genes', defined as genes in the MIDAS reference database that are present in at least 90% of samples within a given cohort. Within-host polymorphism rates were computed in core genes.

Orthologous genes present in multiple species can result in read 'stealing' and read 'donating' to species from which the reads did not originate. Thus, we excluded a set of genes belonging to a 'blacklist' composed of genes present in multiple species. This blacklist was constructed in *Garud et al., 2019*, using USEARCH (*Edgar, 2010*) to cluster all genes in human-associated reference genomes with a 95% nucleotide identity threshold. Since some genes may be absent from the MIDAS

database but may nevertheless be shared across species, we implemented another filter (as in *Garud et al., 2019*) in which genes with $c \geq 3$ in at least one sample in our cohort were excluded from analysis of polymorphism rate or gene changes over time.

## Inferring SNVs within bacterial species

To call SNVs, we mapped reads to a single representative reference genome as per the default MIDAS software. Reads were mapped with Bowtie2, with default MIDAS mapping thresholds: global alignment, MAPID ≥94.0%, READQ ≥20, ALN_COV ≥0.75, and MAPQ ≥20. Species were excluded from further analysis if reads mapped to ≤40% of their genome. We additionally excluded samples from further analysis if they had low median read coverage ($\underline{D}$) at protein coding sites. Specifically, samples with $\underline{D} < 5$ across all protein coding sites with nonzero coverage were excluded. This MIDAS SNV output was then used for computing within-species polymorphism rates and inferring the number of strains present for each species in each sample (see below).

To compute polymorphism rates, additional bioinformatic filters were imposed to avoid read stealing and donating across different species. First, we did not call SNVs in blacklisted genes present in multiple species. Additionally, we excluded sites in a given sample if $D<0.3\,\underline{D}$ or $D>3\,\underline{D}$ as these sites harbor anomalously low or high coverage compared to the genome-wide average $\underline{D}$ . Additional filters are described below.

## Shannon diversity, species richness, and polymorphism rate calculations

Shannon diversity and richness were computed within each sample by including any species with abundance greater than zero. Rarefied species richness estimates are based on HMP1-2 samples rarefied to 20 million reads and Poyet samples rarefied to 5 million reads. SNV and gene content variation within a focal species were ascertained only from the full dataset and not the rarefied dataset.

The polymorphism rate of a species in a sample was computed as the proportion of synonymous sites in core genes with intermediate allele frequencies (0.2 ≤$f$ ≤0.8), as was done in *Garud et al., 2019*. Only species with a MIDAS marker gene coverage of 10 or more in 10 or more samples were included, yielding 69 species in 249 HMP stool donors and 15 species in four *Poyet et al., 2019* donors. As explained in SI text 1 in *Garud et al., 2019*, this is quantitatively similar to the more traditional population genetic measure of heterozygosity, $H=E[2f(1 - f)]$, in which intermediate frequency alleles contribute the most weight to heterozygosity. By computing polymorphism with the criteria 0.2 ≤$f$ ≤0.8, we avoid inclusion of low-frequency sequencing errors, which can otherwise greatly influence the mean heterozygosity. Polymorphism rates were computed separately for synonymous (fourfold degenerate) and nonsynonymous (onefold degenerate) sites. The degeneracy of sites was determined based on MIDAS output.

## Temporal changes in polymorphism rates and gene content

Polymorphism change was computed as the difference in polymorphism rates between time points within a host. Gene gains and losses between time points were computed in species with sufficient prevalence (at least 10 samples with marker gene coverage of at least 10, as in the polymorphism analysis above) by identifying genes with copy number $c \leq 0.05$ (indicating gene absence) in one sample and $0.6 \leq c \leq 1.2$ (with marker coverage ≥20×) in another (indicating single copy gene presence). These thresholds were used in *Garud et al., 2019*, when inferring gene changes in temporal data and reflect a range of copy numbers expected in either the absence of a gene or presence of a single copy of a gene given typical coverage values in growing cells (*Korem et al., 2015*). These copy number cutoffs were chosen to avoid spuriously analyzing genes linked to multiple species. In such cases, mapping artifacts in which reads can be arbitrarily assigned to multiple species cannot be disentangled. For example, a gene present in multiple species would likely have copy number significantly deviating from 1 (including values that lie in an ambiguous zone of 0.05–0.6, as well as >>1), reflecting the joint abundances of the multiple species. Thus, although we may miss many biologically interesting multi-copy genes (e.g. transporter genes in *Bacteroides*; *Wexler and Goodman, 2017*), our thresholds avoid confounding our analysis with read stealing or donating among different species. Filters for coverage and blacklisted genes were applied as described above.

## Strain number inference

We used StrainFinder (*Smillie et al., 2018*) to infer the number of strains present within each species in each HMP1-2 metagenomic sample. To do so, we used allele frequencies from MIDAS SNV output, generated as described above. For each species in each host, all multi-allelic sites with coverage of 20× or greater were passed as input to StrainFinder. Species/host pairs which had fewer than 100 sites with 20× coverage were removed from the analysis. StrainFinder was then run on each sample separately for strain numbers 1, 2, 3, and 4, and the optimal strain number was chosen based on the Bayesian information criterion. This range of strain number was chosen for biological reasons. A number of studies have demonstrated that at most a small handful of strains (between 1 and 4) not sharing a common ancestor within the host are ever observed within a single gut microbiome at any one time (*Garud et al., 2019*; *Truong et al., 2017*; *Verster et al., 2017*; *Yassour et al., 2018*). Additionally, for the four densely longitudinally sampled hosts in *Poyet et al., 2019*, multiple analyses employing distinct sequencing strategies and strain phasing techniques have similarly concluded that a maximum of four strains were present at any one time within a host for the ~30 most prevalent species (*Poyet et al., 2019*; *Wolff et al., 2021*; *Zheng et al., 2022*). Thus, four strains were chosen as the maximum to accommodate the range of observed possibilities.

## Statistical analyses

### Model construction and evaluation

Using data from the HMP and *Poyet et al., 2019*, we examined the relationship between within-species genetic diversity and the gut microbiome community diversity. Within-species genetic diversity was estimated with polymorphism rate and strain richness. Community diversity was estimated with the Shannon index, species richness estimated on the whole data, and species richness calculated on the data rarefied to an equal number of reads per sample (as described above). Generalized additive mixed models (mgcv function from the mgcv R package – RStudio version 1.2.5042) were used for most analyses, except when the response data were counts, such as the number of strains, gene gains, or gene losses. In these cases, we used GLMMs (glmmTMB function from the glmmTMB R package – RStudio version 1.2.5042). GLMMs are currently more flexible than GAMs in the range of count models that it can fit (https://bbolker.github.io/mixedmodels-misc/glmmFAQ.html; *Bolker, 2023*). glmmTMB can deal with overdispersion in count data via two versions of negative binomial distributions negative binomial1 and negative binomial2, respectively, with linear and quadratic parameterization (*Hardin and Hilbe, 2018*), and can handle zero-truncated count data (*Shonkwiler, 2016*) with truncated Poisson and truncated negative binomial for both linear and quadratic parameterizations. In our case, strain count is an overdispersed positive variable, so a zero-truncated distribution was needed. We fit three different GLMMs with truncated-Poisson, truncated-negative binomial1 and truncated-negative binomial2, and then selected the best model based on the Akaike information criterion (AIC) as described in *Brooks et al., 2017*. The same methods were used to fit GLMMs for gene gains and losses.

To account for variation in sequencing depth, which can affect estimates of both community diversity and within-species genetic diversity, we added read count per sample (coverage) as a covariate to all generalized mixed models. Species name, subject identifier, and sample identifier were added as random effects to account for variation between different species and subjects, and to account for non-independence between observations. The R syntax and statistics of all generalized models are reported in *Supplementary file 2*.

In GLMMs, the predictors were standardized to zero mean and unit variance before analyses. We first assessed random effects significance by comparing nested models where each random effect was dropped one at a time using the LRT (ANOVA function from the R stats package) and only significant random effects were included in the final models. We then assessed the fixed effects' significance with LRTs implemented in the drop1 function in the R stats package. This function drops individual terms from the full model and reports the AIC and the LRT p-value. All the p-values reported for the GLMMs correspond to LRT and not to the Wald p-values reported by glmm.summary function from the R package glmmTMB, as was recommended in https://bbolker.github.io/mixedmodels-misc/glmmFAQ.html; (*Bolker, 2023*). We again used LRTs to compare the full significant models to null models including all random effects but no fixed effects other than the intercept. The difference in Akaike information criterion (ΔAIC) between full and null model and their associated p-values are

reported in *Supplementary file 1e–g*. As an additional evaluation of the goodness of fits, we estimated the coefficient of determination ($R^2$) using the r2 function from the performance R package. Two values are reported: the marginal $R^2$, a measure of the variance explained only by fixed effects, and the conditional $R^2$, a measure of the variance explained by the entire model.

We evaluated model fits by inspecting the residuals using the DHARMa library in R (simulateResiduals and plot functions) for the GLMMs and by inspecting residual distributions and fitted-observed value plots using the gam.check function from the mgcv R package for the GAMs. Adjusted $R^2$ values (from gam.summary function from the mgcv R package) are reported as a goodness of fit for the GAMs. All model outputs (summary function from mgcv and glmmTMB R packages) are reported in the *Supplementary file 2*.

To study the relationship between focal species polymorphism and community diversity calculated at higher taxonomic ranks (from genus to phylum), we used GTDBK and the Genome Taxonomy Database (GTDB) (*Chaumeil et al., 2019*) to annotate MIDAS reference genomes. Richness at each level was estimated with the total number of distinct taxonomic units in the sample. The Shannon index was calculated based on the relative abundances table from MIDAS: at each taxonomic level, we used the sum of the abundances of all species belonging to that taxonomic level to calculate the Shannon index (using the diversity function from the R vegan library). We then fit two separate GAMs for each taxonomic rank (from genus to phylum) with either Shannon diversity or richness as the predictors of within-species polymorphism rate (with the coverage per sample as a covariate and species name, sample and subject identifiers as random effects). These GAMs were fitted with a beta error distribution with logit-link function, chosen because polymorphism rate is a continuous value strictly bounded by 1, and all the terms were smoothed terms (see *Supplementary file 1c* and *Supplementary file 2* sections 1–3 for additional model details).

We repeated the same methods for focal species synonymous and nonsynonymous polymorphism separately. See *Supplementary file 1b and d* and *Supplementary file 2* section 4–6 for details of the models applied to nonsynonymous polymorphism.

## Analysis of strain counts per focal species

To study the relationship between community diversity and the number of strains within a focal species in the HMP, we restricted the analysis to 184 focal species genomes with at least 100 nucleotide sites with 20× coverage in a sample. We fit separate GLMMs with strain count in a focal species as a function of community diversity estimated with Shannon diversity, species richness, or rarefied species richness. Since strain number is positive count data, we compared alternative zero-truncated count models based on the AIC score (AICtab function from bbmle R library) (*Brooks et al., 2017*). We fit the model with the truncated negative binomial distribution (truncated_nbinom2 or truncated_nbinom1 in glmmTMB; the second best fit) in order to resolve the overdispersion detected in the best fit (the truncated Poisson model). Overdispersion was tested using the check_overdispersion function from the performance R package as described here: https://bbolker.github.io/mixedmodels-misc/glmmFAQ.html; (*Bolker, 2023*).

As described above for focal species polymorphism, we tested the relationship between focal species strain count and community diversity at higher taxonomic levels from genus to phylum, fitting a separate GLMM with strain count in a focal species as a function of each metric of diversity (Shannon and richness) at higher taxonomic levels (from genus to phylum). All GLMM details are reported in *Supplementary file 1f* and *Supplementary file 2* sections 7–9.

## Analysis of time series data

To test the predictions of DBD over time, we used HMP samples with multiple time points from the same person to look at the relationship between within-species polymorphism change, defined as the difference between polymorphism rate at two time points, and community diversity at the earlier time point. We fit GAMs with log-transformed polymorphism change as a function of community diversity at the earlier time point, and added the coverage per sample at the earlier time point as a covariate as well as species name, sample, and subject identifiers as random effects (*Supplementary file 2* section 10.1).

In addition, we investigated the effect of community diversity at one time point on gene content variation (gains and losses considered separately) at the subsequent time point. Gene gains and losses

were both overdispersed count data, so we selected the best negative binomial model (between linear and quadratic parameterization) based on the AIC, and fit separate negative binomial GLMMs with gene gain as the response and each of the metrics of community diversity as the predictor, with the same covariates and random effects used in the previous models (*Supplementary file 2* section 10.2). The same method was used to test how gene loss was related to community diversity (*Supplementary file 1g*, *Supplementary file 2* section 10.3).

HMP longitudinal data consisted of hosts sampled at a time lag of ~6 months. To assess the relationship between within-species genetic diversity and community diversity at higher temporal resolution, we used the same methods to analyze longitudinal metagenomic data from four more frequently sampled healthy stool donors (hosts *am*, *an*, *ao*, and *ae*) (*Poyet et al., 2019*). Stool from donor *am* was sequenced over 18 months with a median of 1 day between samples; *an* over 12 months (median 2 days between samples); *ao* over 5 months (median 1 day between samples); and *ae* over 7 months (median 2 days between samples). We looked at polymorphism change and gene gains and losses between two time points in the 15 species with a minimal marker gene coverage of 10 in at least 10 samples. Community diversity was estimated with Shannon diversity (unrarefied) and richness calculated on rarefied data to 5 million reads per sample.

We used the same methods as in HMP time series to study the relationship between community diversity at the initial time point and polymorphism change between the initial time point and all the future time points. We fit Gaussian generalized additive mixed models with log-transformed polymorphism change as the response and the interaction between community diversity at the first time point and the number of days between time points as the predictor. Covariates included coverage, species name, sample, and subject identifiers as random effects (*Supplementary file 1h*, *Supplementary file 2* sections 11.1 and 11.2). To study the relationship between gene variation (gains and losses separately) and diversity at the first time point, we fit negative binomial GLMMs with gene variation as a function of the interaction between diversity at the first time point and the number of days between the two time points, with the same covariate and random effects as used above for polymorphism change over time (*Supplementary file 1i*, *Supplementary file 2* sections 11.3–11.6).

## Acknowledgements

We sincerely thank members of the Garud and Shapiro labs, and Pleuni Pennings, for their feedback during the development of this paper. NRG received support from the Paul Allen Frontiers Group, a University of California Hellman fellowship, a UCLA Faculty Career Development award, and the Research Corporation for Science Advancement. DWC received funding support from NIH R25 MH 109172. BJS was supported by a Natural Sciences and Engineering Research Council of Canada (NSERC) Discovery Grant and a Canada Research Chair. We also thank Djordje Bajić and two anonymous reviewers for their constructive suggestions that substantially improved the manuscript.

## Additional information

### Funding

| Funder | Grant reference number | Author |
| --- | --- | --- |
| Allen Institute | Paul Allen Frontiers Group | Nandita R Garud |
| Research Corporation for Science Advancement | | Nandita R Garud |
| Natural Sciences and Engineering Research Council of Canada | | B Jesse Shapiro |
| Canada Research Chairs | | B Jesse Shapiro |
| National Institutes of Health | R25 MH 109172 | Daisy Chen |

The funders had no role in study design, data collection and interpretation, or the decision to submit the work for publication.

## Author contributions
Naïma Madi, Conceptualization, Software, Formal analysis, Investigation, Visualization, Methodology, Writing – original draft, Writing – review and editing; Daisy Chen, Data curation, Formal analysis, Methodology, Writing – review and editing; Richard Wolff, Formal analysis, Investigation, Methodology, Writing – review and editing; B Jesse Shapiro, Conceptualization, Supervision, Funding acquisition, Investigation, Visualization, Writing – original draft, Project administration, Writing – review and editing; Nandita R Garud, Conceptualization, Data curation, Supervision, Funding acquisition, Investigation, Methodology, Writing – original draft, Project administration, Writing – review and editing

## Author ORCIDs
Naïma Madi (iD) http://orcid.org/0000-0001-8793-7389
Daisy Chen (iD) http://orcid.org/0000-0001-6516-7029
B Jesse Shapiro (iD) http://orcid.org/0000-0001-6819-8699
Nandita R Garud (iD) http://orcid.org/0000-0003-4217-4407

## Ethics
Human subjects: All human-derived samples used in this study were previously published. We include no additional identifiable or sensitive information.

## Decision letter and Author response
Decision letter https://doi.org/10.7554/eLife.78530.sa1
Author response https://doi.org/10.7554/eLife.78530.sa2

# Additional files

## Supplementary files
- Supplementary file 1. Supplementary tables a–i.
- Supplementary file 2. R syntax and statistics of all generalized models.
- MDAR checklist

## Data availability
The raw sequencing reads for the metagenomic samples used in this study were downloaded from *Human Microbiome Project Consortium, 2012* and *Lloyd-Price et al., 2017* (URL: https://aws.amazon.com/datasets/human-microbiome-project/); and *Poyet et al., 2019* (NCBI accession number PRJNA544527). All computer code for this paper is available at https://github.com/Naima16/DBD_in_gut_microbiome, (copy archived at swh:1:rev:eef9dc1643f279e4bb2f6d7e19c4fd7d5acc384f; *Madi et al., 2023*).

The following previously published datasets were used:

| Author(s) | Year | Dataset title | Dataset URL | Database and Identifier |
|---|---|---|---|---|
| Human Microbiome Project | 2013 | Human Microbiome Project Data Set | https://aws.amazon.com/datasets/human-microbiome-project/ | Registry of Open Data on AWS, human-microbiome-project |
| Poyet M, Groussin M, Gibbons SM, Avila-Pacheco J, Jiang X, Kearney SM, Perrotta AR, Berdy B, Zhao S, Lieberman TD, Swanson PK, Smith M, Roesemann S, Alexander JE, Rich SA, Livny J, Vlamakis H, Clish C, Bullock K, Deik A, Scott J, Pierce KA, Xavier RJ, Alm EJ | 2019 | BIO-ML: The Broad Institute-OpenBiome Microbiome Library enables mechanistic studies with isolates, genomes, and longitudinal metagenomics and metabolomic data | https://www.ncbi.nlm.nih.gov/bioproject/PRJNA544527 | NCBI BioProject, PRJNA544527 |

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
