## [Editor Report]

This paper analyses meta-genomic human gut microbiome data to understand how biodiversity arises and can be maintained. It makes an important contribution by strengthening the diversity-begets-diversity hypothesis and linking it to signatures of gene loss expected from the Black Queen hypothesis. While only correlative data is used to draw conclusions, the methods are solid and alternative hypotheses are clearly outlined.

---

## [Decision Letter]

**Decision letter after peer review:**

Thank you for submitting your article "Community diversity is associated with intra-species genetic diversity and gene loss in the human gut microbiome" for consideration by *eLife*. Your article has been reviewed by 3 peer reviewers, and the evaluation has been overseen by a Reviewing Editor and Detlef Weigel as the Senior Editor. The following individual involved in review of your submission has agreed to reveal their identity: Djordje Bajić (Reviewer #1).

Essential revisions:

Three main weaknesses of the work were pointed out that would require extensive work to address, which would be beyond the scope of the current paper. The reviewers are requesting instead that these weaknesses be made very clear in the manuscript. The first is that DBD is only one possible process that can explain the observed patterns, and alternative hypotheses are not sufficiently outlined. The second is that the sequencing depth of the analyzed samples is little for gut microbiomes and hence allows only to look at a fraction of the strain-level diversity present. Third, it is not clear what the mechanism would link the BQH and DBD. Based on these points, it would be important to

(i) tone down the conclusions,

(ii) elaborate on certain concepts,

(iii) discuss these limitations of your study, and

(iv) consider whether any further analyses of your data might address these points (more detail below).

*Reviewer #2 (Recommendations for the authors):*

The study is purely correlative, some of the correlations are weak or not significant, only a subset of the analysed species shows a significant positive correlation, and when changing the method to assess the diversity not the same species are being identified to have a positive correlation (diversity-SNV analysis) or opposite trends are being observed (diversity-strain number). This means that the detected patterns are not very robust. The authors refer to a recent pre-print (Estrela et al) that provides experimental support for DBD. However, the environmental conditions in that study were very different from the one found in the gut, in that only a single carbon source was provided. DBD is only one possible process that may explain the observed patterns (see above). Therefore, I recommend the authors tune down their conclusions, discuss their findings more critically, i.e. offer alternative explanations, or argue better why these alternative explanations are less likely than DBD to explain the observed patterns.

From the discussion, it is not clear what would explain the DBD at finer taxonomic scale. It would be great to have a bit more insights about how the authors believe that diversity at e.g. the species, genus, or even family level could influence strain-level diversity within an individual species. What type of niches can be created by the presence of additional families? And how can these niches be occupied by distinct strains of the same species? Although it is true that some strains may be ecologically distinct, I would still assume that most strains occupy similar niches and are competing for similar nutrients. Moreover, wouldn't we expect that there is so much functional redundancy in the gut microbiome already that a slight increase in taxonomic diversity would not necessarily create new metabolic niches from which closely related strains with similar metabolic capacities would profit? I think it would help to explain a bit more how the DBD should work out at the strain level and to what extent there is evidence that the increased strain-level diversity can really be adaptation to new niches created by other microbes in the system.

The HMP1-1 and Poyet samples were rarified to 20 and 5 mio reads. This seems very little sequencing depth considering the high amount of genus/species level diversity in the human gut microbiome and the fact that the authors want to look at strain-level diversity. I have my doubts that this amount of data will allow to quantitatively assess diversity at the strain level in these samples, which may in part explain the not very robust correlations observed. How much of the total diversity is assessed with this number of reads? Rarefaction curves of SNVs or genes per species discovered when sub-setting the datasets across samples would be helpful. In addition to the low overall sequencing depth, only polymorphisms with a frequency of 0.2-0.8 were considered, which further limits the number of strains that can be detected. What was the idea behind applying this cut-off?

Why was only one species analysed across the entire Poyet dataset? If this is a general pattern it should be observable for more species than just B. vulgatus. While I acknowledge that read coverage may not be high enough for all samples for other species, there should be enough consecutive timepoints with sufficient coverage for some species. Unless there is a good reason why such analysis can only be done for B. vulgatus, I strongly recommend to extend this analysis to other community members to find further support for DBD-like patterns.

Along the same lines, the Poyet dataset would offer an opportunity to follow how diversity of B. vulgatus changes over more timepoints than just two in response to the species/genus/family diversity (e.g. over the entire 18 months?). This could provide interesting new insights that may help to understand how diversity changes over time at different levels. Why not look at changes in diversity across all timepoints? Would we expect that DBD continues over a longer period of time or would we see some type of negative feedback, because of ecological control kicking in at one point leading to oscillations of diversity?

The figures with the correlations should be improved. Specifically, Figure 2, 3, and 5 include too many data points of different species on top of each other. It is impossible to look at the distribution of the data of individual species and appreciate the existence of correlations. In panel A of Figure 2, I see only one data point for Dialister invisus. Why? Is the color legend missing next to panel B in Figure 3?

For the StrainFinder analysis, it was assumed that species for which no site passed the 20x threshold are presented by a single strain. This seems wrong in my opinion. Such data should be excluded as the shallow sequencing simply does not allow assessing the number of strains of that species in these samples.

Out of 68 species only 15 or 18 show a significant slope. This means that there are more species that display either EC or no correlation (as EC may not always detectable). This takes away from the conclusion that just DBD explains the patterns of diversity found in the gut microbiome, and should be acknowledged.

Line 148: What is the overlap of the 15 and 18 detected positive correlations between Shannon and richness analysis? From comparing the legends of Figure 2A and B, it seems that the overlap is not great. It would be helpful if the authors could state the overlap in their paper and discuss it.

Line 147: Significance of the correlations should be corrected for multiple testing. Would the identified correlations still be significant?

How does the method to identify gene loss/gene gain differentiate between gene loss and strain-level dynamics? That is, we do not know whether a certain genome lost genes or a strain lacking those genes happened to dominate the community. If such strain happened to migrate into the community for example, this would not be evidence for the BQH.

*Reviewer #3 (Recommendations for the authors):*

The paper is very well written and generally well-integrated with previous work by the group and by others.

I found the main novelty of the present work to be the usage of time series data to enquire about how present microbiome community diversity may influence within species polymorphism at a future time point, motivated by the mechanism underlying the Black Queen Hypothesis (BQH) put forward ten years ago.

There are however several points that I think need to be revised:

1. It was not clear to me in the authors' introduction and discussion if and how the DBD hypothesis is integrated with the BQH.

Do the authors consider these two hypotheses to be independent? What sort of mechanisms do the authors envisage to drive a positive association between community diversity and polymorphism? And is the mechanism underlying the BQH assumed to result in the fixation of a gene loss within the focal species or may it also result in polymorphism as in Morris, Papoulis and Lenski 2014 Coexistence of evolving bacteria stabilized by a shared black queen function?

I think clarifying these points would strengthen the manuscript.

2. Another issue that was not clear to me was why the authors compute the polymorphism rates with only synonymous sites. If there is a reason to exclude non-synonymous sites it should be mentioned in the manuscript. In addition, the abstract and the conclusions should precisely state that the significant correlations are with polymorphism rates at synonymous sites.

3. The manuscript emphasizes the finding of a positive correlation in several species but does not emphasize that for the majority of species no correlation was found. Thus, the conclusion that DBD prevails (pg. 7 line 150 and abstract) looks a bit exaggerated.

---

## [Author Response]

Essential revisions:Three main weaknesses of the work were pointed out that would require extensive work to address, which would be beyond the scope of the current paper. The reviewers are requesting instead that these weaknesses be made very clear in the manuscript. The first is that DBD is only one possible process that can explain the observed patterns, and alternative hypotheses are not sufficiently outlined. The second is that the sequencing depth of the analyzed samples is little for gut microbiomes and hence allows only to look at a fraction of the strain-level diversity present. Third, it is not clear what the mechanism would link the BQH and DBD. Based on these points, it would be important to(i) tone down the conclusions,(ii) elaborate on certain concepts,(iii) discuss these limitations of your study, and(iv) consider whether any further analyses of your data might address these points (more detail below).

We thank the reviewers and editors for their thorough and thoughtful assessment of our paper and we have worked to address these essential revisions. First, we have extensively re-written this manuscript to be more balanced in evaluating alternative hypotheses to DBD, including ecological controls (EC) and abiotic drivers of diversity. Second, we clarify that our results apply only to relatively abundant species with sufficient sequencing depth to study sub-species diversity. We explicitly state that further work will be required to extend these results to rarer species (please see Discussion). Third, we acknowledge it is not clear which mechanism would link DBD and BQH – or if these models operate orthogonally. This is also an avenue for future work, which we mention in the Discussion.

As suggested, we:

i) Tone down our conclusions and mention alternative explanations for the observed patterns;

ii) Elaborate on certain concepts, including why sub-species genetic diversity might alterniches and thus be impacted by community diversity, and also how DBD or EC could arise by de novo mutation and/or strain migration across hosts;

iii) Include several paragraph of caveats and limitations of our study (please see Discussion); and

iv) We comprehensively analyze 15 species from four hosts instead of only Bacteroides vulgatus from a single host from Poyet et al. to reinforce the correlation between community diversity and gene loss over time. We also add new analyses to determine whether the DBD relationship differs between non-synonymous and synonymous nucleotide variants. Finally, we closely evaluated the pathway-level analysis and found that the gene-level analyses were more robust. We have therefore removed the pathway analyses.

Below we elaborate on these points and include passages from the revised text.

Reviewer #2 (Recommendations for the authors):The study is purely correlative, some of the correlations are weak or not significant, only a subset of the analysed species shows a significant positive correlation, and when changing the method to assess the diversity not the same species are being identified to have a positive correlation (diversity-SNV analysis) or opposite trends are being observed (diversity-strain number). This means that the detected patterns are not very robust. The authors refer to a recent pre-print (Estrela et al) that provides experimental support for DBD. However, the environmental conditions in that study were very different from the one found in the gut, in that only a single carbon source was provided. DBD is only one possible process that may explain the observed patterns (see above). Therefore, I recommend the authors tune down their conclusions, discuss their findings more critically, i.e. offer alternative explanations, or argue better why these alternative explanations are less likely than DBD to explain the observed patterns.

While we still think that DBD provides a plausible mechanism to explain the observed patterns, we are now careful to acknowledge alternative mechanisms. We more prominently list possible caveats and limitations in the Discussion. We also give more weight in the Results to contrasting patterns that we observe in the data. For example:

“The negative strain number-richness relationship also held at all other taxonomic ranks (GLMM, *P*<0.05, LRT) (Figure S3, Table S6, Supplementary File 1 section 9), while the strain number-Shannon diversity relationship was generally positive (Figure S3, Supplementary File 1 section 8). These effects also appear to be species-specific: for example, the number of *Bacteroides vulgatus* strains per host is positively correlated with both Shannon diversity and richness (consistent with DBD predictions) whereas *B. ovatus* has no relationship with Shannon diversity but a negative correlation with richness (consistent with EC; Figure 2A, B). Together, these results reveal that different components of community diversity can have contrasting effects on the diversity slope.”

From the discussion, it is not clear what would explain the DBD at finer taxonomic scale. It would be great to have a bit more insights about how the authors believe that diversity at e.g. the species, genus, or even family level could influence strain-level diversity within an individual species. What type of niches can be created by the presence of additional families? And how can these niches be occupied by distinct strains of the same species? Although it is true that some strains may be ecologically distinct, I would still assume that most strains occupy similar niches and are competing for similar nutrients. Moreover, wouldn't we expect that there is so much functional redundancy in the gut microbiome already that a slight increase in taxonomic diversity would not necessarily create new metabolic niches from which closely related strains with similar metabolic capacities would profit? I think it would help to explain a bit more how the DBD should work out at the strain level and to what extent there is evidence that the increased strain-level diversity can really be adaptation to new niches created by other microbes in the system.

Thank you for raising this important point. We have now added the following Introduction paragraph to outline the reasons we might expect strain-level diversity to affect niche preferences, and thus be relevant to DBD:

“Such fine-scale strain-level variation has important functional and ecological consequences; among other things, strains are known to engage in interactions that cannot be predicted from their species identity alone (Goyal et al., 2022). Although closely-related bacteria are expected to have broadly similar niche preferences, finer-scale niches may differ below the species level (Martiny et al., 2015). For example, the acquisition of a carbohydrate-active enzyme by *Bacteroides plebeius* allows it to exploit a new dietary niche in the guts of people consuming nori (seaweed) (Hehemann et al., 2010), and single nucleotide adaptations permit *Enterococcus gallinarum* translocation across the intestinal barrier resulting in inflammation (Yang et al., 2022). Despite their potential phenotypic effects, it is unknown if such fine-scale genetic changes are favored by higher community diversity (due for example to niche construction, as predicted by DBD) or suppressed (due to competition for limited niche space, as predicted by EC). Competition could also lead to DBD if focal species evolve new niche preferences to avoid extinction (Mitri and Foster, 2013; Schluter, 2000) – an idea with some support in experimental microcosms (Meyer and Kassen, 2007) but largely unexplored in natural communities.”

Regarding functional redundancy, we have added the following Discussion paragraph:

“Our findings in human gut metagenomes are compatible with the BQH under the assumption that increasing community diversity also increases the availability of leaky gene products – which may not be the case if genomes in the gut microbiome are functionally redundant, as inferred in a recent study (Tian et al., 2020). This study found that species in the gut microbiome were highly redundant at the level of annotated metabolic pathways (KEGG orthologs) and that more functionally redundant microbiomes were more resistant to colonization by fecal transplants. Relatively low-redundancy microbiomes could therefore be more easily colonized, but might also require migrants to encode more gene functions in order to persist. Importantly, functional redundancy may be high at the level of well-annotated metabolic functions, but low at the finer level of individual gene families, as demonstrated in marine microbiomes (Galand et al., 2018) but not yet studied explicitly in the gut. Here we report that genome reduction in the gut is higher in more diverse gut communities. This could be due to de novo gene loss, preferential establishment of migrant strains encoding fewer genes, or a combination of the two. The mechanisms underlying this correlation remain unclear and could be due to biotic interactions – including metabolic cross-feeding as posited by some models (Estrela et al., 2022; San Roman and Wagner, 2021, 2018) but not others (Good and Rosenfeld, 2022) – or due to unknown abiotic drivers of both community diversity and gene loss.”

The HMP1-1 and Poyet samples were rarified to 20 and 5 mio reads. This seems very little sequencing depth considering the high amount of genus/species level diversity in the human gut microbiome and the fact that the authors want to look at strain-level diversity. I have my doubts that this amount of data will allow to quantitatively assess diversity at the strain level in these samples, which may in part explain the not very robust correlations observed. How much of the total diversity is assessed with this number of reads? Rarefaction curves of SNVs or genes per species discovered when sub-setting the datasets across samples would be helpful. In addition to the low overall sequencing depth, only polymorphisms with a frequency of 0.2-0.8 were considered, which further limits the number of strains that can be detected. What was the idea behind applying this cut-off?

We thank the reviewer for their concerns about potentially under-calling SNVs and genes when subsetting datasets. However, we only rarefied the data when ascertaining species richness, since richness is most affected by sampling effort. We realize that it may have been confusing as to when rarefaction was applied to the data and therefore have modified the Methods to be clearer about our approach as follows:

“SNV and gene content variation within a focal species were ascertained only from the full dataset and not the rarefied dataset.”

As mentioned above, we now clearly mention the caveat that our results apply only to relatively abundant focal species for which we are able to estimate sub-species diversity. For example, Strain Finder only estimates the number of strains within species with sufficient coverage. We also include sequence coverage (number of total reads in the metagenome) in all of our GLMM and GAM analyses, so the effects of sampling effort are taken into account throughout.

We thank the reviewer for asking about the 0.2-0.8 thresholds, as it provides us an opportunity to clarify our methodology. First we wish to emphasize that these thresholds of 0.2 – 0.8 were not used for strain detection. Rather, all SNVs were used for strain detection with StrainFinder, provided that the site had 20x or greater coverage. We clarify this in the main text as well as the methods.

The 0.2-0.8 threshold was applied when computing polymorphism rates, as in Garud, Good et al. 2019, where we analyzed the same HMP dataset. This measure of polymorphism rate is similar to a more traditional measure of heterozygosity H=E[2f(1−f)], in which intermediate frequency alleles contribute the most weight. Our approach is preferable, however, because it is more robust to low-frequency sequencing errors that can overwhelm the average in H. We clarify our citations to our previous paper in the Methods section, where our rationale is explained.

Why was only one species analysed across the entire Poyet dataset? If this is a general pattern it should be observable for more species than just B. vulgatus. While I acknowledge that read coverage may not be high enough for all samples for other species, there should be enough consecutive timepoints with sufficient coverage for some species. Unless there is a good reason why such analysis can only be done for B. vulgatus, I strongly recommend to extend this analysis to other community members to find further support for DBD-like patterns.

We now expand the analysis to 15 species and 4 individuals as we elaborate in our response above.

Along the same lines, the Poyet dataset would offer an opportunity to follow how diversity of B. vulgatus changes over more timepoints than just two in response to the species/genus/family diversity (e.g. over the entire 18 months?). This could provide interesting new insights that may help to understand how diversity changes over time at different levels. Why not look at changes in diversity across all timepoints? Would we expect that DBD continues over a longer period of time or would we see some type of negative feedback, because of ecological control kicking in at one point leading to oscillations of diversity?

Thank you for this very useful suggestion. As described above, the analysis of these time series now includes time lag in the model. We find that the positive correlation between community diversity and focal species polymorphism at a future time point holds on shorter time lags, then flattens and becomes negative at longer time lags (Figure 5 and S4). The relationship between community diversity and gene loss is always positive, with slight variation depending on the time lag. It is intriguing to think about possible oscillations of DBD and EC, and we hope this can be explored in future work. We now mention this in the first Discussion paragraph:

“Based on the time series metagenomic data analyzed here, the predictions of DBD also tend to hold over short time scales, but fail over longer time scales of several months. Whether this leads to a terminal plateau of diversity, or whether ecological disturbances lead to cycles of DBD and EC, deserves further study.”

The figures with the correlations should be improved. Specifically, Figure 2, 3, and 5 include too many data points of different species on top of each other. It is impossible to look at the distribution of the data of individual species and appreciate the existence of correlations. In panel A of Figure 2, I see only one data point for Dialister invisus. Why? Is the color legend missing next to panel B in Figure 3?

We agree that the original figures were difficult to read and we have remade them completely. The new figures 2, 3, and 4 now show each of the nine most prevalent focal species as a separate panel, to avoid overlapping points, and also show the GLMM or GAM model fits in lower panels to illustrate the broad patterns supported by all the data.

For the StrainFinder analysis, it was assumed that species for which no site passed the 20x threshold are presented by a single strain. This seems wrong in my opinion. Such data should be excluded as the shallow sequencing simply does not allow assessing the number of strains of that species in these samples.

We thank the reviewer for this suggestion, and we have incorporated it into our revised analysis. Now, for the purposes of strain finding, we consider absent from a given sample any species in which there are fewer than 100 polymorphic sites with greater than 20x coverage. 9% of species/sample pairs used in the original analysis were excluded after applying this cutoff.

100 sites was chosen as a conservative threshold for strain detection. Prior work has shown that when multiple strains of a species are present within a single host, fixed genetic differences between strains tend to greatly outnumber genetic differences between individuals belonging to the same strain (Yassour et al. 2016; Garud, Good et al. 2019; Zheng et al. 2022). In particular, while tens of thousands of sites or more typically segregate between strains, only hundreds to a few thousands of sites typically segregate within a strain. Additionally, assuming a conservative sequencing error rate of 10^-3 per base pair, thousands of sites may be spuriously marked as polymorphic due to technical error—such polymorphisms will, however, be found at low frequency, and are explicitly accounted for by StrainFinder. In total, genetic differences between strains should account for a large proportion of the total polymorphisms present within a sample when multiple strains coexist. Therefore, we expect that a significant majority of the (minimum) 100 polymorphic sites will be informative as to the presence of multiple strains.

Out of 68 species only 15 or 18 show a significant slope. This means that there are more species that display either EC or no correlation (as EC may not always detectable). This takes away from the conclusion that just DBD explains the patterns of diversity found in the gut microbiome, and should be acknowledged.

Our original intention in reporting these simple linear correlations for each species was to give illustrative examples, not to establish statistical significance. We have now removed these individual correlations for each species, which were not corrected for multiple tests, nor did they include important covariates like coverage. We now focus entirely on the more robust GLMM or GAMs, which include these covariates and provide better statistical descriptions of the entire dataset in question.

Line 148: What is the overlap of the 15 and 18 detected positive correlations between Shannon and richness analysis? From comparing the legends of Figure 2A and B, it seems that the overlap is not great. It would be helpful if the authors could state the overlap in their paper and discuss it.

As mentioned in the response above, these individual linear correlations have been removed.

Line 147: Significance of the correlations should be corrected for multiple testing. Would the identified correlations still be significant?

As mentioned above, these correlations were meant to be anecdotal and illustrative. Since they are not statistically robust, we have removed them in favor of the more appropriate GLMMs and GAMs. This obviates the need for multiple test corrections, since the entire dataset is used to test a single overarching hypothesis.

How does the method to identify gene loss/gene gain differentiate between gene loss and strain-level dynamics? That is, we do not know whether a certain genome lost genes or a strain lacking those genes happened to dominate the community. If such strain happened to migrate into the community for example, this would not be evidence for the BQH.

Indeed, our method does not distinguish between these possibilities, and we now clarify this as follows in the Results:

“Our method does not explicitly distinguish between gene gain/loss arising from recombination or deletion versus replacement of strains with different gene content.”

However, we disagree with the reviewer that gene loss due to strain dynamics rather than de novo gene loss would not be evidence for BQH. We think that selection for strains encoding fewer genes in more diverse communities (with the proviso that these more diverse communities encode more public goods) is also compatible with the general predictions of BQH. This is now mentioned in the Discussion as follows:

“Here we report that genome reduction in the gut is higher in more diverse gut communities. This could be due to de novo gene loss, preferential establishment of migrant strains encoding fewer genes, or a combination of the two.”

Reviewer #3 (Recommendations for the authors):The paper is very well written and generally well-integrated with previous work by the group and by others.I found the main novelty of the present work to be the usage of time series data to enquire about how present microbiome community diversity may influence within species polymorphism at a future time point, motivated by the mechanism underlying the Black Queen Hypothesis (BQH) put forward ten years ago.

Thank you for these kind words.

There are however several points that I think need to be revised:1. It was not clear to me in the authors' introduction and discussion if and how the DBD hypothesis is integrated with the BQH.Do the authors consider these two hypotheses to be independent? What sort of mechanisms do the authors envisage to drive a positive association between community diversity and polymorphism? And is the mechanism underlying the BQH assumed to result in the fixation of a gene loss within the focal species or may it also result in polymorphism as in Morris, Papoulis and Lenski 2014 Coexistence of evolving bacteria stabilized by a shared black queen function?I think clarifying these points would strengthen the manuscript.

We thank the reviewer for their thoughtful comments. In the Discussion, we now elaborate further on how our results are consistent with DBD and BQH. However, we fully acknowledge that future work is needed to integrate DBD and BQH into a single unified model. We have now written a new final Discussion paragraph on this topic:

“In summary, we demonstrate how metagenomic data can be used to test the predictions of eco-evolutionary theory, including DBD, EC, and the BQH. It remains to be seen whether the distinct eco-evolutionary processes proposed by DBD and the BQH operate orthogonally or whether they interact. If BQH leads to gene losses that remain polymorphic rather than being lost entirely from the species (Morris et al., 2014) – or invasions of strains with fewer genes that remain incomplete and do not replace the resident strain – this could be viewed as a form of diversification and perhaps a special case of DBD. Here we considered gene loss as a directional process; we did not attempt to distinguish between directional changes in gene copy number and the complete extinction of a gene, which is difficult to show using metagenomic data. Future work could attempt to resolve this point and to potentially combine DBD and BQH into a unified theory.”

Additionally, we appreciate the reviewer mentioning the possibility that BQH could lead to polymorphism within a species. We acknowledge this could be true especially if multiple strains of a species colonize a host. In the Discussion we now mention this possibility needs to be investigated further:

“Finally, we measured community diversity from the phylum to the species level, not below. We therefore did not investigate how the BQH could extend to maintain gene content variation within a species, as has been shown experimentally in *E. coli* (Morris et al., 2014). This could be an avenue for future work.”

2. Another issue that was not clear to me was why the authors compute the polymorphism rates with only synonymous sites. If there is a reason to exclude non-synonymous sites it should be mentioned in the manuscript. In addition, the abstract and the conclusions should precisely state that the significant correlations are with polymorphism rates at synonymous sites.

We thank the reviewer for this comment. We have now added an analysis of nonsynonymous sites, which largely follow the same patterns observed for synonymous sites. This can be seen by comparing Figures S1 (synonymous) and S2 (non-synonymous), which both show generally positive associations between community diversity and focal species polymorphism. The trends are very similar for both synonymous and non-synonymous polymorphism, although generally less statistically significant for the non-synonymous sites. This is to be expected since non-synonymous sites are less numerous, providing a smaller sample size that could reduce statistical power. This is now reported in the Results, and in the Discussion as follows:

“Nonsynonymous variation also tended to track positively with both measures of community diversity but was only statistically significantly associated with phylum and class richness. This suggests that evolutionarily older, less selectively constrained synonymous mutations and more recent nonsynonymous mutations that affect protein structure both track similarly with measures of community diversity. Nonetheless, a parsimonious explanation for possible differences between the two classes is that while they are affected similarly, we have more statistical power to identify correlations in the more numerous synonymous mutations. This merits further investigation.”

3. The manuscript emphasizes the finding of a positive correlation in several species but does not emphasize that for the majority of species no correlation was found. Thus, the conclusion that DBD prevails (pg. 7 line 150 and abstract) looks a bit exaggerated.

The simple linear correlations for each species were meant to be illustrative, but were not properly representative of statistical significance. We have now removed these individual correlations for each species, which were not corrected for multiple tests, nor did they include important covariates like coverage. We now focus on the more robust GLMM or GAMs, which include these covariates and provide better statistical descriptions of the entire dataset in question. This now shows that a positive relationship between community diversity and focal species polymorphism is the predominant pattern, although of course this varies by time scale and by species. We make this clear in the Results in this passage, among others:

“These effects also appear to be species-specific: for example, the number of *Bacteroides vulgatus* strains per host is positively correlated with both Shannon diversity and richness (consistent with DBD predictions) whereas *B. ovatus* has no relationship with Shannon diversity but a negative correlation with richness (consistent with EC; Figure 2A, B). Together, these results reveal that different components of community diversity can have contrasting effects on the diversity slope.”

We have now toned down the conclusion that “DBD prevails” and have rewritten the Abstract to include the following text, which we think is more balanced:

“We find that both intra-species polymorphism and strain number are positively correlated with community Shannon diversity. This trend is consistent with DBD, although we cannot exclude abiotic drivers of diversity. Shannon diversity is also predictive of increases in polymorphism over time scales up to ~4-6 months, after which the diversity slope flattens and then becomes negative—consistent with DBD eventually giving way to EC. Also supporting a complex mixture of DBD and EC, the number of strains per focal species is positively associated with Shannon diversity but negatively associated with richness.”